# What happens when pharmacist independent prescribers lead on medicine management in older people's care homes: a qualitative study

Linda Birt  ,[1,2] Lindsay Dalgarno,[3,4] Fiona Poland,[5] David Wright,[1,2] Christine Bond[4]

¹Faculty of Medicine and Health Sciences, University of East Anglia, Norwich, UK
²School Healthcare, University of Leicester, Leicester, UK
³School of Health and Wellbeing, University of Glasgow, Glasgow, UK
⁴School of Medicine, Medical Sciences and Nutrition, University of Aberdeen, Aberdeen, UK
⁵School of Health Sciences, University of East Anglia, Norwich, UK

**Correspondence to**
Linda Birt; linda.birt@uea.ac.uk

## ABSTRACT

**Objective** Older people in care homes frequently experience polypharmacy, increasing the likelihood of medicine-related burden. Pharmacists working within multidisciplinary primary care teams are ideally placed to lead on medication reviews. A randomised controlled trial placed pharmacists, with independent prescribing rights (PIPs), into older people care homes. In the intervention service, PIPs worked with general practitioners (GPs) and care home staff for 6 months, to optimise medicine management at individual resident and care home level. PIP activity included stopping medicines that were no longer needed or where potential harms outweighed benefits. This analysis of qualitative data examines health and social care stakeholders' perceptions of how the service impacted on care home medicine procedures and resident well-being.

**Design** Pragmatic research design with secondary analysis of interviews.

**Setting** Primary care pharmacist intervention in older people care homes in England, Scotland and Northern Ireland.

**Participants** Recruited from intervention arm of the trial: PIPs (n=14), GPs (n=8), care home managers (n=9) and care home staff (n=6).

**Results** There were resonances between different participant groups about potential benefits to care home residents of a medicine service provided by PIPs. There were small differences in perceptions about changes related to communication between professionals. Results are reported through three themes (1) 'It's a natural fit'—pharmacists undertaking medication review in care homes fitted within multidisciplinary care; (2) 'The resident is cared for'—there were subjective improvements in residents' well-being; (3) 'Moving from "firefighting" to effective systems'—there was evidence of changes to care home medicine procedures.

**Conclusion** This study suggests that pharmacist independent prescribers in primary care working within the multidisciplinary team can manage care home residents' medicines leading to subjective improvements in residents' well-being and medicine management procedures. Care home staff appreciated contact with a dedicated person in the GP practice.

**Trial registration** ISRCTN 17847169

## STRENGTHS AND LIMITATIONS OF THIS STUDY

⇒ The study draws on data from a relevant sample of 38 health and social care professionals involved in medicine management in care homes.
⇒ The analytical approach foregrounded the effect of pharmacist-led medication review on the older resident in care homes, therefore having relevance to practices supporting safer medicines in this group of people.
⇒ A limitation is the absence of residents' voices, due to recruitment challenges with this group.
⇒ The study provides insights into activity provided by specialist pharmacists with independent prescribing qualifications so it cannot be assumed that more generalist pharmacists would have the experience to make specialised medicine management decisions.

## INTRODUCTION

Older people in care homes (CHs) are often subject to complex medicine regimes which may include the concurrent administration of more than five different medicines daily.[1 2] This is known as polypharmacy; it is associated with a higher risk of adverse side effects including falls, hospitalisation and mortality.[3 4] The complexity of managing and administering multiple medicines increases CH staff workload.[5] This can lead to increased rates of medicine administration errors, and while few errors result in serious consequences for residents, all need to be reported and followed up.[6] The Care Quality Commission highlights management of medicines as an area of concern and holds it under constant review.[7] In UK CHs, prescribing and medication reviews for individual residents are the responsibility of clinical professionals in the primary care team.[8] A Structured Medication Review (SMR) is recommended for all CH residents.[9] An SMR takes a personalised approach, drawing on

shared decision-making principles to increase safety and efficacy.[9] Pharmacists are being routinely employed to lead on SMRs in primary care practices, and an English government initiative to enable pharmacists to have a role within CHs is gathering momentum.[10] Pharmacists have a specialised knowledge of medicine-related burden[11] including 'drug burden' where the cumulative effect of the active ingredients of medicines can impact on physical and cognitive function.[12] If pharmacists have the postgraduate Independent Prescriber qualification, they can start (prescribe) or stop (deprescribe) medicines within their area of clinical competence.[13 14] This enhanced role allows them to use their expert clinical knowledge of medicines to deliver services in CHs.

This paper reports on a secondary analysis of interview data collected from pharmacists with independent prescribing rights (PIPs), general practitioners (GPs) and CH staff as part of the process evaluation[15] undertaken as part of the Care Home Independent Prescribing Pharmacist Study (CHIPPS).[16] Within the trial, a service intervention was designed which enabled pharmacist-led medicine management for CHs. PIPs worked within CHs on a weekly basis over 6 months to review their medicine systems and residents' personal medicines.[17] PIPs were based within the general practice aligned to the CH. The secondary analysis aimed to understand how the PIP role was enacted and PIPs, GPs and CH staffs' perceptions of how PIP activity impacted on CH medicine procedures and resident well-being, concepts which were outside the remit of the trial process evaluation.[17]

## METHODS
### The CHIPPS study
The pharmacist-led service offered in CHs is briefly described below to set the findings of the secondary analysis in the practice context; full details of the trial design are available at references[16 17] and trial outcomes are available.[18]

The pharmacist-led service was delivered through a triad of a GP, a PIP based at the GP practice, and up to 24 CH residents, from up to three CHs, registered with the GP. Service specifications were developed following focus groups with stakeholders to guide the PIPs' work (see online supplemental file 1). The primary trial outcome was reduction in falls for residents.[18] However, in a prestudy focus group, health and social care professionals (HSCPs) wondered whether there might be wider benefits for residents.[19] The communication between the triad, their understanding of responsibilities of PIPs and outcomes for residents are the focus of this secondary analysis of process evaluation interview data. Within the trial, 25 triads were recruited to the intervention arm. Sixteen PIPs were already working with the GP; the remaining nine were allocated a GP practice for the purposes of the study. PIPs undertook a 2 day bespoke face-to-face training programme, produced a portfolio of evidence of competencies and finally were signed off to prescribe in

older people medicine by an independent GP.[20] To spread researcher workload (triad recruitment, data collection), the intervention was delivered over four 6 month phases between January 2018 and March 2020. Data collection for the process evaluation occurred between May 2019 and March 2020 as each phase completed. The secondary analysis of interview data was carried out by LB and LD.

### Design
Pragmatism is a research methodology suited to characterising knowledge about how behaviours and actions impact on healthcare systems and outcomes. This approach can help appreciate the value of knowledge within its context of activity, uses and how these relate to the experience of addressing practical problems.[21] As this work aimed to understand and learn from the diverse ways different stakeholders engaged with and perceived the intervention, a pragmatic design provided a way to recognise multiple realities of people involved.

### Recruitment and sample
During the process evaluation, health and social care stakeholders in each triad where the intervention was delivered, were invited to take part in a semistructured interview at the end of each 6 month phase: PIPs n=23 (2 withdrew and contact details not available), GPs n=25 and CH managers n=38. A reminder invitation email was sent after 2 weeks. CH managers provided recruitment information to staff, residents and relatives. Those interested in participating returned expressions of interest to study researchers. A purposive sampling framework was designed to ensure representation across stakeholder roles, location and phases of the intervention. However, response rates were low and therefore all those replying were invited to interview.

### Data collection
The process evaluation semistructured interview guide drew on outcomes from the earlier feasibility study[22] and reflected the Medical Research Council process evaluation domains.[23] Questions explored participants' experiences of the service intervention, anticipated and unanticipated outcomes, including multidisciplinary communication and outcomes for residents (see online supplemental file 2). Interviews were undertaken by LB and LD academic researchers, between May 2019 and March 2020, either in person or by telephone and audio recorded and professionally transcribed. Informed consent was obtained before interview.

### Data management and analysis
An inductive thematic analysis was used to characterise participants' experiences of delivering or receiving the pharmacist-led service. Thematic analysis provides a structured, methodical way for the researcher to familiarise with the data and to organise, analyse and report findings; importantly it is not strongly aligned to any epistemological stance,[24] so was appropriate within the pragmatic design. Both researchers familiarised themselves

with the data through checking transcripts for accuracy, then reading and making analytical memos.[25] Next, they independently developed open codes (see online supplemental file 3). The developing framework was discussed with the wider research team to refine the boundaries of the codes. Each interview was coded and any uncertainty arising during coding was discussed between researchers (see online supplemental file 4). While the researchers predominantly coded interviews they had carried out, each coded two of the other's interviews to check for consistency of approach. This along with fortnightly review meetings increased dependability during analysis. The coding framework was revised iteratively as analyses progressed, for example grouping similar codes into a category of meaning. Categories were refined and, following discussions with the research team and patient and public partners, the final themes (overarching, abstracted semantic interpretations of the data[26]) were developed. While a theme does not rely on specific quantities of data, here each theme provided extensive examples of data. To further support trustworthiness in results, illustrative examples of data within the theme were shared at research meetings to seek consensus from healthcare professionals, including pharmacists and GPs. Participant validation was not undertaken due to the time between interviews conducted in each phase and the end of study analysis.

## Patient and public involvement

Public involvement was supported by the Patient and Public Involvement in Research group (further details at https://nspccro.nihr.ac.uk/working-with-us/public-patient-and-carer-voice-in-research). Public members were involved at all stages of the trial and in relation to this paper, they reviewed data and provided comments on emerging interpretations. Public members had experience of having relatives receiving polypharmacy and of working in CHs; this enabled the research team to have confidence in their interpretations and to consider in greater detail potential resident-focused benefits.

## RESULTS

Interviews, lasting between 30 and 90 min, were undertaken with 38 HCPs: 14 PIPs, nine GPs, nine CH managers and six CH staff. Participants were recruited from 18 of the 25 triads: six in Scotland, four in Northern Ireland and eight in England. There was no expression of interest from any stakeholder in seven triads. In three of these triads, the PIPs did not deliver the intervention; in the remaining four triads, the demographics and trial outcomes were similar to the sample interviewed. Data were collected from a heterogeneous sample of professionals and locations as shown in online supplemental file 5 (demographic characteristics of interview participants). Data were collected from triads with older people CHs who provide personal care and some social activities (residential) and dual registration care homes which provide personal and nursing care. We identify illustrative quotes within quotation marks. For substantial quotes, we provide professional role and unique triad number; non-italicised text in quotes is for explanation.

Three themes were developed. Theme (1) *'It's a natural fit'—multidisciplinary working in CHs* explores GP, PIP and CH staff differing experiences of this new role; (2) *'The resident is cared for'—shared goals in medicine management* provides exemplars of improvement in residents' experience of medication and the impact of targeted deprescribing (3) *'Moving from "firefighting" to effective systems'* reports on the improvement to organisation systems across CHs and GP practices and whether this lasted beyond the intervention. Participants mostly recounted positive experiences but some identified challenges inherent in pharmacist-led medicine management in CHs.

## Theme 1: 'It's a natural fit'—multidisciplinary working in care homes

This theme reports GP, CH staff and PIPs' experiences of pharmacist-led medicine management. The GP and CH staff valued the PIPs' activity for differing reasons, suggesting that a PIP role can meet both sets of needs.

### GP perceptions

GPs highlighted the safety advantage of having PIPs within the multidisciplinary team, explaining that it was helpful to have 'another pair of eyes' to increase patient safety:

> Having a pharmacist who had good knowledge of all kinds of medications, going through polypharmacy, with a fine-tooth comb and picking up any errors or things we could do better. GP_21.

GPs also drew on PIPs' expertise of medicine administration for the wider benefit of CH residents, *'things that can be crushed or can be opened or whether they need to be changed to suspensions'*, explaining the PIP *'seems to know the answer for everything with regards to medication interactions and probably the thing I ask her most is appliances, different catheters and things'* GP_8.

GPs explained that as PIPs could spend several hours within the home with a small number of residents, they could understand residents' medicine needs in greater detail, *'it has brought up things that I might not have thought about'* GP_16. GPs and care home staff were generally open to suggestions from PIPs. However, two PIPs not based in the GP practice prior to the intervention struggled to have their prescribing role accepted, *'I couldn't just go off and prescribe things because the doctor would need to know … it was quite obvious she didn't want me to do anything without putting it through her'* PIP_19. This may suggest a need for actively reviewing and clarifying accountability of roles so that PIPs may be fully integrated into the multidisciplinary team.

While GPs found it hard to quantify the impact of the intervention on their workload, on reflection they mostly described how they were now dealing with fewer daily

enquiries from CHs. Where PIPs and GPs had established communication channels, the PIPs regularly identified residents and issues that the GP should look at in their weekly ward round:

> It has reduced the time taken to see patients as I can be confident their meds are all up to date, tests required for routine monitoring have been flagged up and I have been able to action these GP_3.

Not all GPs agreed that the PIP intervention was a 'natural fit', rather one GP described a situation where they believed the intervention had increased their workload as the PIP had requested biochemical monitoring which was '*creating work…actual fact I didn't change resident's dose, there was no indication to do so and I did wonder what I was trying to achieve.*' GP_19. In part, this situation arose as the PIP did not have full access to the resident's clinical records and was unaware a blood test had been taken during a recent hospital visit, highlighting the importance of PIPs being able to have full access to the primary care team's systems and patient records.

### Care home team perceptions
CH managers and staff identified benefits in having a dedicated pharmacist attached to the home. Most CH teams explained they had gained improved accessibility and advice by having PIPs as a point of contact in the primary care team, '*if you're phoning a surgery about a pharmacy issue, probably better speaking to a pharmacist than a GP*' CH manager_21a.

Where multidisciplinary working was embedded, CH teams explained the clinical skills of each professional were respected, and teamwork supported effective resident care:

> The [CH] nurses' assessments are being taken as a valuable tool, because the nurses are observing, the nurses are giving the assessment, and the nurses are liaising with the PIP, to prescribe what they think is needed, so I think it's a win, win situation, the nurses are feeling valued, and the PIP as well CH manager_21b.

However, in two homes, PIPs were not able to become integrated into the CH. In one, the PIP reported that CH staff were unclear of their role. In another, there was confusion about the legal position of PIPs to prescribe and the manager insisted the GP signed off all the PIP's prescribing activity. This highlights the importance of understanding and trust across the triad.

### PIP perceptions
PIPs commonly reported that their prescribing authority meant decisions and resident care could be followed up in a timely way:

> I think the benefits of having pharmacists in Primary Care is we can go in and make the changes and own it and follow it up rather than pass it on, …leaving it for other people to follow up PIP_17.

A few PIPs mentioned the role of the pharmacy technician as potentially supporting their work, as pharmacy technicians would undertake stock control and check medicine administration records for accuracy with stock. Only one PIP appeared to have worked with a technician during the intervention '*…a [pharmacy] technician that's employed to go around the homes, I had a relationship with them already, … a lot of the work that they had already done, …so it meant me going in wasn't as huge job to turn around the whole care home*' PIP_14. A GP pointed out that it may not be good use of a pharmacist '*to be going in and counting tablets and working out what the home needed*' GP_21.

In summary, GPs and PIPs made clear what expertise an independent prescribing pharmacist could bring to the multidisciplinary team and saw a continuing place for pharmacists in CHs. While most CH teams valued the PIP role, a few were still uncertain on the range of their specific responsibilities.

### Theme 2: 'The resident is cared for'—shared goals in medicine management
GPs and PIPs highlighted a strength of having a PIP as 'the linchpin in medicine management' in that they had detailed pharmaceutical knowledge allowing them to take actions which would directly and specifically benefit residents. PIPs reviewed prescriptions and could actively seek to rectify any prescribing errors and reduce the drug burden related to adverse effects from medicines. By doing this, they importantly improved safety for the resident and helped to make MAR (Medication Administration Record) sheets fit for purpose thereby potentially reducing the risk of administration error. Changes PIPs could make ranged from small actions such as changing the time of eye drop administration, so residents were not woken every night by staff, changing tablet preparations to soluble form, to complex titrating of medicines, for example, reducing doses of morphine-based painkillers and stopping antipsychotic medicines. PIPs reviewed all medicines in the context of a resident's current biomedical markers and stage of life.

Residents, alongside some members of the multidisciplinary team, were sometimes uncertain of the PIP's credentials, and PIPs reported that a few residents were reluctant to have their medicines reviewed, preferring to remain with prescriptions given by the GP or secondary care team. There were a few triads where CH teams struggled to recall changes which had impacted on residents' well-being. These proved to be either homes where PIPs struggled to build working relationships and therefore their medication activity was negligible, or homes that already had highly developed relationships with either the PIP allocated to the intervention or with another pharmacist. In this latter case it was reported that most residents already received regular medication reviews.

### Managing resident prescriptions
Independent prescribing activity was a key role of PIPs in the multidisciplinary team. Their involvement could

directly benefit residents, often because this could progress a new prescription much more speedily:

> If my residents are feeling their symptoms, then refer to the PIP, and the PIP will prescribe immediately, there will immediate response, and then the residents will be happy CH manager_21b.

Other CH teams identified how having direct contact with a PIP stopped conditions worsening, '*somebody having dry skin, and being able to contact a pharmacist, definitely improved care and stopped things becoming more acute, skin breaking down*' CH Manager_16. They explained that for such conditions, they would not have contacted the GP but would have waited for the weekly GP visit, with the ensuing delay potentially leading to the worsening of conditions.

Key to successful deprescribing, reducing or stopping a medicine, was the PIP having trustful relationships with the CH team, the GP and the resident or their family. The importance of this trustful relationship was made evident when CH teams explained how they were now confident in trying to titrate prescriptions down as they knew the PIP was easy to contact and had agreed that if there were adverse effects the process could be stopped, and the medicine reinstated:

> Seeing PIP on a weekly basis meant we were able to make some huge reductions but also we were both very honest with each other so when we had got rid of almost everything and then resident's behaviour got worse again, we were able to add in a tiny little bit of something and I mean tiny bit which was appropriate CH Manager_9.

Titrating antipsychotic medicine is a complex process and each stakeholder needs to be fully committed and in good communication with each other. Here, a PIP explains how the process continued after the intervention ended, illustrating the continued benefit of having PIPs working in the GP practice:

> We had started to titrate it down and for a few days they were okay and then they started to need a lot of diazepam so we increased it back up but now it has come back down by half; they are checking and watching even though the study has ended because I have been involved and generally keeping an eye on things too they still have my contacts. PIP_1.

Discussions on deprescribing were also enabled between PIPs and the resident or their relative, thereby providing more patient-centred care. A CH manager explains:

> …talking through that with the PIP and the resident she has come full circle and is off her Butec patches with no pain; that was agreed with her and her family … she didn't mind coming off and giving it a trial period CH Manager_6.

### Shared decisions

While it was intended that PIPs would be actively engaged with residents as part of the intervention, frequently residents did not have the mental capacity for this. However, even though a PIP stated that residents would possibly have little recall of their conversation, they still found it relationally valuable to meet with residents, '*I enjoyed meeting them all and then they become more than just a name on a page, you remember the person behind the medication*' PIP_1. A few PIPs also reported the benefit of engaging with family:

> I could really get on well with the family, she wanted to know everything what I was going to do so that was more interesting because I could discuss things, a really good example of joint working with the family PIP_16.

In summary, most participants readily identified ways in which the intervention had directly benefited individual residents. The key factor enabling PIPs to work in ways which benefited residents stemmed from the trust the CH team, resident and their family could place on the PIP as a readily accessible, clinically competent person who could react appropriately if adverse consequences arose to any medicine change.

### Theme 3: 'Moving from "firefighting" to effective systems'

Within the CH, most of the PIPs actively reviewed and where appropriate improved medicine ordering and administration systems, for example streamlining dispensing systems, consolidating MAR charts and stock control.

### Streamlining medicine supply and stock control

PIPs could articulate their detailed knowledge of the role of the community pharmacist and the processes which may be in place when a medicine is started or stopped, particularly the time it can take for automated systems to register the change. This 'insider knowledge' placed PIPs in an ideal position to educate other professionals and try to streamline medicine ordering processes:

> Geriatrician didn't realise that although it was stopped in the MAR chart the next month's is out there in the van waiting to be delivered, just anticipating those fires PIP_9.

PIPs explained that such lack of understanding about the medicine supply process led to stopped medicines still being dispensed and administered and this could be a patient safety issue. The majority reported developing enhanced relationships with community pharmacists and CH staff. If the PIP worked in the GP practice, these relationships were often sustained postintervention, '*any query we just email the PIP we get information get straight from the horse's mouth*' CH Manager_14a.

A few PIPs explained that taking part in the intervention had prompted them to review systems in the GP surgery:

We have 5 people—pharmacy technician, pharmacy assistants, generating prescriptions, dealing with questions, and discharge letters, so one of them is now appointed for the nursing home patients. If we need to discuss anything, she's the one that helps if we make a change, she'll change it on their file, and then she'll liaise as well with the nursing home PIP_22.

## Rationalising record and stock

Working alongside CH staff PIPs allocated time to review MAR charts and consider if medicines could move to PRN (as required) or homely medicines. The staff reported they appreciated dedicated time to discuss things and simplified MAR charts:

It sorted out things that we didn't need any longer and put things a bit more into place, like the PRNs particularly where some people didn't need it on their MAR charts any longer, … I suppose that had been overlooked which does happen I'm afraid, yes it was useful CH staff_19.

Removing no-longer-needed prescriptions and identifying changes in the ordering systems helped reduce workload, '*it really reduced our workload, in terms of kind of managing medication in the nursing home*' CH manager_11.

Many of the PIPs explained that by regularly attending the care homes, they were able to identify waste and unused stock and could take measures to reduce waste:

I spoke a few times with the dispensing pharmacists about errors that came up on the MAR chart etc. we tried to address those issues. I spoke to the appliance contractors not just the Pharmacy, … I had to address it in a really decisive manner PIP_16.

One GP highlighted the importance of cost-saving inherent in stock control, '*I wouldn't have had a clue about it* {over stock} *because I don't go into the store cupboards so that was a massive cost saving*' GP_16.

In summary, many PIPs were found to take an active lead in reviewing and advising on more efficient medicine systems. However, their efforts to improve systems could be thwarted if either the CH staff or GP team were unwilling, or unable, to adopt new ways of working.

## DISCUSSION

This secondary analysis of interview data from the process evaluation of the CHIPPS study[15] specifically focused on how the PIP role was enacted and PIPs, GPs and CH staffs' perceptions of the impact of PIP activity on CH medicine procedures and resident well-being. The results provide contextualised understandings of the independent prescribing pharmacist role within the wider primary care team, with most PIPs and GPs suggesting PIPs should have a continued role in management of CH medicine systems and residents' medicines. Importantly,

most participants reported perceptible benefits from the PIP intervention whether through improved resident well-being or increased safety and streamlining of CH procedures. Benefits from the intervention appeared greater when there was professional trust between GP, CH staff and the PIP. Examining exceptions can help better understand what resources, training or other preparation might need to be put in place for stakeholders to benefit from extended pharmacist roles.

GPs and PIPs made clear a pharmacist independent prescriber could bring specialised clinical knowledge to the multidisciplinary team and they saw a continuing place for pharmacists in CHs. While most CH teams valued the role of the PIP, a few were still uncertain on the range of their specific responsibilities. Pharmacists have an increasingly valued place within multidisciplinary primary care teams, and this is evident within policy statements.[8 27] However, importantly, the pharmacist role needs to cover more than reviewing medicines to maximise impact. As part of a multidisciplinary team, pharmacists can provide dedicated specialist input into SMR[9] by drawing on their clinical pharmaceutical knowledge.[28] A pharmacist independent prescriber may be able to relieve the GP of some prescribing activity and so help to backfill GP shortages as fewer European GPs are now recruited.[29] Yet as the CHIPPS study found, there are differences in the character of pharmacist medication review[15] and there are suggestions that there may need to be quality check tools on practice.[30]

The review of individual resident's medicines undertaken as part of this intervention was reported by CH staff and PIPs as having readily demonstrated benefits for residents' subjective quality of life. This understanding is important as the evidence on objective improvements in resident outcomes, that is, admission to hospital, falls or mortality from this trial and others is inconclusive.[18 31 32] Holland *et al*,[33] suggested that measures relating to older people's quality of life may be a better measure. This is borne out in trails that fail to find significant change in primary outcomes such as falls but do find positive outcomes in reduction of drug-related burden.[18 34] Our study suggests that measures of impact may need to be related to residents' everyday experiences, such as better engagement with others, more alert at mealtimes, rather than clinical risk of side effects from multiple no-longer-needed medicines. We found that there needed to be trust between the CH staff and PIPs in order for some of the pharmacists' recommendations to be enacted; CH staff needed to have confidence the pharmacist would respond quickly if a resident's behaviour became distressed and a challenge to manage.

As part of the intervention service specification, PIPs spent time actively reviewing, and where appropriate, improving CH medicine systems. For example, reviewing and improving dispensing systems, consolidating MAR charts and monitoring stock control. These were reported as having benefits for CH staff in that administration and reordering were easier, and potentially safer. Improved

management of stock reduced wastage and therefore had financial benefits. These findings resonate with evidence supporting the place of pharmacists within CHs.[35] Nonetheless, future work might explore if this is optimum use of a pharmacist's skills or if this might be a technician-level activity.

The exceptions to the positive results reported so far occurred when relational factors inhibited the scope of pharmacist contributions. For example, when the relationship between pharmacist and GP or CH could not be successfully established, or understanding of the pharmacist's legal clinical scope was not understood. This suggests the need for strategies which will develop shared understanding of the potential of each other's roles. Our results resonate with a survey by Kahn *et al*[36] that aimed to understand factors impacting on interprofessional working in primary care; they found that the time the team had worked together, and opportunities for formal and informal communication were important in enabling the team to develop trust and collaborative working practices. In our study, the few PIPs who had not previously worked in the trial GP surgery reported greater resistance to their role and made fewer substantive changes to residents' medicines and CH medicine ordering and administration systems. It is important that all stakeholders understand the pharmacist role so as to support collaborative working practices.[37]

### Strengths and limitations
Missing from our study are the resident and relatives' opinions of medication review and the place of the pharmacist in this. We attempted to recruit the residents consented to the intervention arm of the trial through CH staff. It is unclear why this was unsuccessful with only three residents expressing an interest, but in part it is likely to be due to the severe cognitive impairment many residents were living with, meaning staff may not have actively encouraged residents or their families to take part in a process evaluation interview. A further point to note is that the sample, while appearing representative of the main trial sample, consists of those who volunteered for the additional process evaluation interview so may not represent the views of those who declined. This point along with the limited number of participants represented in a small number of codes means that the transferability of the data needs to be considered with caution as the results provide insights and understandings from a very specific group of HSCPs involved in supporting older resident CHs. All pharmacists were independent prescribers, and all had received additional training in medicine for older people. It cannot be assumed that more generalist pharmacists in a GP surgery would have the experience to make the specialised medicine management decisions demonstrated by these pharmacist independent prescribers who had received additional training. There are recommendations to make CH pharmacists a designated specialty.[38] However, further training may be needed as inexperienced pharmacists may rely on template-driven reviews.[39]

The bespoke training developed for this intervention appeared appropriate for enhancing pharmacists' clinical skills and confidence.[40] Pharmacists in our study had medication review responsibilities only for between 9 and 24 residents, so allowing them dedicated time to come to understand those residents' individual needs.

The CHIPPS RCT found the fall rate risk ratio for the intervention group compared with the control group was not significant. However, the Drug Burden Index outcome significantly favoured the intervention.[18] Reduction in drug related problems is reported in another pharmacist led review intervention,[41] indicating there is potential for pharmacist led review to have some positive impacts for CH residents relating to side effects of some medicines. Further economic evaluation would be required to examine if this level of pharmacist input would be cost effective in practice. It may be that some tasks could be allocated to the developing role of pharmacy technicians.[42]

There are practice implications for other roles within the wider healthcare team. For example, during interviews, PIPs referred to their work with community pharmacists, describing the community pharmacist position as key in dispensing medicines. Future work exploring medicines management in CHs might include the community pharmacist or dispensing pharmacy so that the efficiency of medicine ordering and dispensing can be further optimised. This might reduce the potential safety risk, reported in this study, which occurs when medicines which have been deprescribed remain on the MAR chart.

The CHIPPS process evaluation was completed in 2020 just as primary care networks (PCN) were being introduced in England and at the start of the COVID-19 pandemic. A key aspect of the role was the presence of the pharmacy within the CH, the move to more 'on-line' working since the COVID-19 pandemic may make this aspect of the intervention more difficult to implement. Within PCNs, pharmacists work across several GP practices rather than being within a single practice.[43] Our finding that PIPs not integrated within the GP team faced the most challenges in developing trustful working relationships, mirrors an evaluation of PCNs published in 2022.[44 45] This evaluation reported the need for change in cultures and practices to support additional clinical roles in a GP practice.[39 40] Those working within a PCN rather than a single GP surgery reported feeling a lack of autonomy, belonging and contribution, reflecting the trial experience of PIPs 'dropped' into GP practices.

### CONCLUSION
Independent prescribing pharmacists can successfully take responsibility for medicine management and SMR for older people in CHs. When pharmacists develop professional trustful relationships with GP colleagues and CH staff, they can independently make changes to medicines which benefit resident well-being. Their expertise in medicine systems including stock control and ordering

enabled them to streamline CH systems with the potential benefit of reducing waste and likelihood of administration errors. The changing landscape of global primary care provision indicates that pharmacists will continue to have a key role in leading management of medicines but that how this happens may require monitoring to enable refinement of the delivery model.

**Acknowledgements** Thank you to residents and their families who agreed to receive the intervention, the pharmacists, GP practices and care homes who delivered the intervention and took part in the process evaluation. We would also like to acknowledge the South Norfolk Clinical Commissioning Group (now Norfolk and Waveney ICB) as the study sponsor and the CHIPPS Study team.

**Contributors** DW, CB and FP conceived the overall research design and provided commentary on the progress of the data collection and analysis. LB and LD undertook qualitative data collection and analysis, all authors reviewed emerging results. LB led on producing the manuscript. All authors commented on versions of the manuscript, and all agree to the final version. DW is guarantor.

**Funding** This is a summary of independent research funded by the National Institute for Health Research (NIHR) under its Programme Grants for Applied Research Programme (Grant Reference Number RP-PG-0613-20007). The views expressed are those of the authors and not necessarily those of the National Health Service, the NIHR or the Department of Health.

**Competing interests** DW received speaker fees from Desitin Pharma and speaker fees and unrestricted education grants from Rosemont Pharmaceuticals. All other authors have no competing/conflict of interest.

**Patient and public involvement** Patients and/or the public were involved in the design, or conduct, or reporting or dissemination plans of this research. Refer to the Methods section for further details.

**Patient consent for publication** Not required.

**Ethics approval** This study involves human participants. English ethical approval was gained from East of England Cambridge Central Research Ethics Committee 17/EE/0360 (28 November 2017; this applied to research in Northern Ireland). Scottish ethical approval was gained from Scotland A research Ethics Committee 17/SS/0118 (7 December 2017). Participants gave informed consent to participate in the study before taking part.

**Provenance and peer review** Not commissioned; externally peer reviewed.

**Data availability statement** Data are available upon reasonable request. The data sets used and/or analysed during the current study are available from the corresponding author on reasonable request.

**ORCID iD**
Linda Birt http://orcid.org/0000-0002-4527-4414

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
