## [Reviewer comments · BMJ Open]

ARTICLE DETAILS

TITLE (PROVISIONAL)	What happens when pharmacist independent prescribers lead on medicine management in older people's care homes: a qualitative study
AUTHORS	Birt, Linda; Dalgarno, Lindsay; Poland, Fiona; Wright, David; Bond, Christine

VERSION 1 – REVIEW

REVIEWER	Lim, Renly University of South Australia Division of Health Sciences, Quality Use of Medicines and Pharmacy Research Centre
REVIEW RETURNED	07-Oct-2022

GENERAL COMMENTS	Thank you for the opportunity to review the paper entitled “A qualitative understanding of outcomes when pharmacist independent prescribers lead on medicine management in older people’s care homes.” for BMJ Open. I have listed some suggestions for your consideration to help improve the manuscript: 1) Title – I wonder if “outcomes” is the appropriate term here. It isn’t clear what outcomes the authors are looking at.2) Abstract - The authors concluded here that the “pharmacist independent prescribers in primary care may be the healthcare professional optimally placed to be the medicines link between general practitioners and care home staff.” I suggest rephrasing the conclusion to more comprehensively reflect the roles and impact of pharmacists independent prescribers. The study found that pharmacists fitted within the multidisciplinary care, led to subjective improvement in residents’ wellbeing and changes in care home medicine procedures. However, the conclusion seems to suggest the ONLY role that pharmacist independent prescribers play is as a “link between general practitioners and care home staff”.3) Introduction - Are all medicines review in the UK normally conducted by GPs? Or only medicine reviews of residents living in care homes? Suggest elaborating a bit more on how care and medicine reviews are provided in the UK (for international readers).4) Introduction - Is the CHIPPS study results published yet? The authors referenced the protocols but not the CHIPPS study results.5) Introduction/methods - The last sentence of paragraph 2 and the whole of paragraph 3 look like methods and would be better placed in the methods section. It describes the methods of the original trial and should form part of methods.6) Results - Can the authors briefly explain what “Dual” and “residential” care homes are, in the methods section?
---

	7) Results - While the authors have categorised the themes rather positively in terms of the pharmacists role, for example Theme 1 "It's a natural fit". However, reading the text on GP perceptions, there were also comments where e.g., GPs were not open to pharmacists prescribing, PIPs increased their workload. I am not sure if "It's a natural fit" was a quote from the interviewees (it isn't explicitly stated) - I would suggest using more neutral themes. 8) Discussion - The first line in the discussion section "This secondary analysis of data from the process evolution of the CHIPPS study" – Similar to my question above, are the process evaluation results published? Or is this the process evaluation results? The way it is written is a bit confusing. 9) Discussion - The lack of involvement of residents is a major study limitation. I wonder if there was any effort made to recruit residents, or whether this was not done because it was perceived as too hard. Looking at the study protocol, the team intended to recruit 880 residents; it seems unrealistic that the authors are not able to recruit even a small sample of residents. Our team recently completed a trial with 248 participants living in residential aged care (nursing home) and it wasn't particularly challenging to recruit residents for qualitative interviews. We interviewed 7 residents (average age 87) mid 2020, at the peak of COVID. The authors also had a "patient and public involvement" section and stated "These enabled the research team to have confidence in their interpretations and to consider in greater detail the RESIDENT FOCUSED BENEFITS." It therefore seems odd that they found recruiting residents for the qualitative interview challenging. This was also not acknowledged in the discussion section, and only mentioned in one sentence under the "strengths and limitations" section. Can the authors comment on this limitation? 10) Under "strengths and limitation of this study", the authors stated that the study had a sample of 37 healthcare professionals. Are care home managers and staff considered healthcare professionals in the UK?
--	---

REVIEWER	Baqir, Wasim Northumbria Healthcare NHS Foundation Trust, Pharmacy
REVIEW RETURNED	16-Oct-2022

GENERAL COMMENTS	Good study and well written. Worth mentioning the Kings Fund report where GPs have said PCN pharmacists create more work for them; perhaps this is because the cohort in this study are more experienced.
---

REVIEWER	Jordan, Susan Swansea University, Nursing
REVIEW RETURNED	27-Nov-2022

GENERAL COMMENTS	The CHIPPS trial represents an important contribution to the 'medicines management' literature - an area vital to patients' wellbeing. This paper is an interesting report on a secondary analysis of interview data collected in conjunction with the large RCT. The description of both positive and negative reactions to the intervention enhances the paper. There should be some indication of how often each sentiment was expressed within the stakeholder group.
---

	Further information on the RCT outcomes and the primary analysis of the interview data would add important context to the interviews presented. The descriptions of subjects, locations and settings for the interviews are too brief. Were the data triangulated with the trial results? What barriers to and facilitators of the pharmacists' roles were identified? The discussion should draw on existing literature, and compare and contrast with other interventions and policy initiatives. Data were collected pre-pandemic: the authors should describe how it is still relevant. The intervention entailed weekly visits to 9-24 residents, sometimes lasting hours, by highly qualified pharmacists. How feasible this would be in locations with fewer resources should be discussed. Many areas, including ours, are experiencing a shortage of pharmacists. As the authors indicate, absence of residents' (and their relatives') views is a major limitation. How did funded patient and public involvement compensate for this? 18 of 25 triads participated. Why did 7 decline the intervention or the interviews? Were the trial outcomes in these 7 any different? Was there any volunteer bias in care home recruitment? E.g. in the deprivation level of the location of the care home? The interview schedule, coding tree, and a sample of text coding should be appended. Minor points p.13, line 3, 'dry skin' should not be described as a 'minor complaint'. The terms 'drug' and 'medication' appear to be used interchangeably. A careful copy edit is needed. Please review absence of possessive apostrophe line 16/17 PIPS or PIPs? Suggest consistency.
--	---

VERSION 1 – AUTHOR RESPONSE

Reviewer: 1

Dr. Renly Lim, University of South Australia Division of Health Sciences

Comments to the Author:

Thank you for the opportunity to review the paper entitled "A qualitative understanding of outcomes when pharmacist independent prescribers lead on medicine management in older people's care homes." for BMJ Open. I have listed some suggestions for your consideration to help improve the manuscript:

1) Title – I wonder if "outcomes" is the appropriate term here. It isn't clear what outcomes the authors are looking at.

1.1 Author response

Thank you for pointing this out we have changed the title to 'What happens when pharmacist independent prescribers lead on medicine management in older people's care homes: a qualitative study?'

2) Abstract - The authors concluded here that the "pharmacist independent prescribers in primary care may be the healthcare professional optimally placed to be the medicines link between general practitioners and care home staff."

I suggest rephrasing the conclusion to more comprehensively reflect the roles and impact of pharmacists independent prescribers. The study found that pharmacists fitted within the multidisciplinary care, led to subjective improvement in residents' wellbeing and changes in care home medicine procedures. However, the conclusion seems to suggest the ONLY role that

pharmacist independent prescribers play is as a “link between general practitioners and care home staff”.

1.2 Author response

The abstract conclusion has been extended to better reflect the range of impacts as below

Conclusion: This study suggests that pharmacist independent prescribers in primary care ~~may be the healthcare professional optimally placed~~ working within the multi-disciplinary team can safely manage care home residents’ medicines leading to subjective improvements in residents’ well-being and medicine management procedures. Care home staff appreciate a dedicated person in the GP practice. ~~to be the medicines link between general practitioners and care home staff~~

3) Introduction - Are all medicines review in the UK normally conducted by GPs? Or only medicine reviews of residents living in care homes? Suggest elaborating a bit more on how care and medicine reviews are provided in the UK (for international readers).

1.3 Author Response

We have deleted the sentence referencing the historical procedure and include a sentence outlining what a structured medication review is with a relevant reference page 4

A Structured Medication Review (SMR) is recommended for all care home residents. (9) An SMR takes a personalised approach drawing on shared decision making principles to increase safety and efficacy ~~Patient medicines review has historically been the role of the general practitioner, other multidisciplinary team members are increasingly undertaking reviews (8).~~ Pharmacists are being routinely employed to lead on ~~medicine review~~

4) Introduction - Is the CHIPPS study results published yet? The authors referenced the protocols but not the CHIPPS study results.

1.4 Author response

We now reference two key study papers:

The main trial paper has just been accepted for publication pending minor amendments (20-Dec-2022 BMJ-2022-071883.R1) The Care Homes Independent Prescribing Pharmacist Study (CHIPPS): A cluster randomised controlled trial to evaluate effectiveness, cost-effectiveness and safety. We will be able to provide full reference in January 2023.

The process evaluation main paper is published. Birt et al (2021). Process evaluation for the Care Homes Independent Pharmacist Prescriber Study (CHIPPS). BMC Health Services Research, 21(1), 1041. <https://doi.org/10.1186/s12913-021-07062-3>

5) Introduction/methods - The last sentence of paragraph 2 and the whole of paragraph 3 look like methods and would be better placed in the methods section. It describes the methods of the original trial and should form part of methods.

1.5 Author responses

We have moved this text to methods section and now include reference to the two published CHIPPS papers

6) Results - Can the authors briefly explain what “Dual” and “residential” care homes are, in the methods section?

1.6 Author response

We have added the following definition. We feel this is better in the results section near the reference to table 1.

Data were collected from triads with older people care homes that provide personal care and some social activities (residential) and dual registration care home which provide personal and nursing care

7) Results - While the authors have categorised the themes rather positively in terms of the pharmacists role, for example Theme 1 “It’s a natural fit”. However, reading the text on GP

perceptions, there were also comments where e.g., GPs were not open to pharmacists prescribing, PIPs increased their workload. I am not sure if “It’s a natural fit” was a quote from the interviewees (it isn’t explicitly stated) - I would suggest using more neutral themes.

1.7 Author response

Thank you for your suggestion rather than change the theme title which was a direct quote from a PIP as are the other two theme headings; we have taken the following steps which we hope will clarify the content.

First before results we have provided better explanation:

We identify illustrative quotes within quotation marks. For substantial quotes we provide professional role and unique triad number; non italicised text in quotes is for explanation.

Second we now draw the reader’s attention to the non-confirming data in this theme:

Page 10 *‘It’s a natural fit’ -multidisciplinary working in care homes* explores GP, PIP and care home staff differing experiences of this new role

Page 11 Not all GPs agreed that the PIP intervention was a ‘natural fit’, rather one GP described a situation where they believed a PIP the intervention had increased their workload as the PIP had requested biochemical monitoring which was *‘creating work...actual fact I didn’t change resident’s dose, there was no indication to do so and I did wonder what I was trying to achieve.’* GP_19.

We also examine the challenges of pharmacist working within GP practices in the discussion.

8) Discussion - The first line in the discussion section “This secondary analysis of data from the process evolution of the CHIPPS study” – Similar to my question above, are the process evaluation results published? Or is this the process evaluation results? The way it is written is a bit confusing.

1.8 Author Response

As above we now include reference to our process evaluation and hope this clarifies the aim of this paper

9) Discussion - The lack of involvement of residents is a major study limitation. I wonder if there was any effort made to recruit residents, or whether this was not done because it was perceived as too hard. Looking at the study protocol, the team intended to recruit 880 residents; it seems unrealistic that the authors are not able to recruit even a small sample of residents. Our team recently completed a trial with 248 participants living in residential aged care (nursing home) and it wasn’t particularly challenging to recruit residents for qualitative interviews. We interviewed 7 residents (average age 87) mid 2020, at the peak of COVID.

The authors also had a “patient and public involvement” section and stated “These enabled the research team to have confidence in their interpretations and to consider in greater detail the RESIDENT FOCUSED BENEFITS.” It therefore seems odd that they found recruiting residents for the qualitative interview challenging.

This was also not acknowledged in the discussion section, and only mentioned in one sentence under the “strengths and limitations” section.

Can the authors comment on this limitation?

1.9 Author Response

We agree the absence of the resident and their relatives voice is a limitation. We now include a sentence to this effect in the discussion

Missing from our study are the resident and their relatives view of medication review and the place of the pharmacist in this. We attempted to recruit the residents consented to the intervention arm of the trial through care home staff. It is unclear why this was unsuccessful with only 3 residents expressing an interest, but in part it is likely to be due to the severe cognitive impairment many residents were living with, meaning staff did not actively encourage residents or their families to take part in the process evaluation interview.

In explanation we tried to gain access to residents and relatives for interview in the process evaluation through the care home manager. This gatekeeper route was not successful. We did not have ethical approval to access personal addresses so could not recruit directly. We had one short interview with a resident. We had three expressions of interest, but this was towards the end of the study and COVID regulations prevented us completing the interviews.

We undertook Patient and public involvement and have added a sentence explaining the characteristic of the public members

Public members had experience of having relatives receiving polypharmacy and of working in care homes; this enabled the research team to have confidence in their interpretations and to consider in greater detail potential resident focused benefits.

10) Under “strengths and limitation of this study”, the authors stated that the study had a sample of 37 healthcare professionals. Are care home managers and staff considered healthcare professionals in the UK?

1.10 Author Response

You are correct care home staff would more usually be referred to as social care staff. In line with English change to terms we now refer to health and social care staff (HSCP)

Reviewer: 2

Dr. Wasim Baqir, Northumbria Healthcare NHS Foundation Trust

Comments to the Author:

Good study and well written. Worth mentioning the Kings Fund report where GPs have said PCN pharmacists create more work for them; perhaps this is because the cohort in this study are more experienced.

2.1 Author Response

Thank you for your comments and suggested additional reference. We agree it is important to acknowledge differing perspective and now reference this report (2022) in the final paragraph of the discussion

Reviewer: 3

Dr. Susan Jordan, Swansea University

Comments to the Author:

1 The CHIPPS trial represents an important contribution to the ‘medicines management’ literature - an area vital to patients’ wellbeing. This paper is an interesting report on a secondary analysis of interview data collected in conjunction with the large RCT. The description of both positive and negative reactions to the intervention enhances the paper. There should be some indication of how often each sentiment was expressed within the stakeholder group.

3.1 Author response

Thank you for comments. We have used terms such as few or many which is standard in thematic analysis reporting. Where it is only a one or two participants, we make this clear by stating one GP etc. We suggest that going further and identifying the number of occasions each point was raised is not required in a thematic analysis.

As suggested below we now include the coding framework and have indicated how many participants interviews were coded to the statement the number of times data were coded. We hope this provides a better sense of how often sentiments were expressed.

2 Further information on the RCT outcomes and the primary analysis of the interview data would add important context to the interviews presented. The descriptions of subjects, locations and settings for the interviews are too brief. Were the data triangulated with the trial results? What barriers to and facilitators of the pharmacists’ roles were identified?

3.2 Author Response

Thank you for your comment we have now referenced the two trial papers in order to provide background and great detail on context.

The main trial paper has just been accepted for publication pending minor amendments (20-Dec-2022 BMJ-2022-071883.R1) The Care Homes Independent Prescribing Pharmacist Study (CHIPPS): A cluster randomised controlled trial to evaluate effectiveness, cost-effectiveness and safety. We will be able to provide full reference in January 2023. Now the trial paper is accepted we have added a line in the discussion that the trial did not report statistical significance in the primary outcome measure of resident fall between intervention and control arms

The process evaluation paper is published. We now referenced this in the manuscript Birt et al (2021). Process evaluation for the Care Homes Independent Pharmacist Prescriber Study (CHIPPS). BMC Health Services Research, 21(1), 1041. <https://doi.org/10.1186/s12913-021-07062-3>. The process evaluation paper contains greater detail on barriers and enablers to pharmacists taking on this role in care homes.

A PhD student undertook further analysis on the deprescribing data from the trial and found that the number of residents and pharmacist-independent prescriber employment within a medical practice were positive predictors of deprescribing. <https://doi.org/10.1111/bcp.15643>

3 The discussion should draw on existing literature, and compare and contrast with other interventions and policy initiatives. Data were collected pre-pandemic: the authors should describe how it is still relevant.

3.3 Author Response

We have amended the discussion to better situate our results within wider literature and policy initiatives and to be more critical of whether this intervention would be implementable in practice on page 16-17

4 The intervention entailed weekly visits to 9-24 residents, sometimes lasting hours, by highly qualified pharmacists. How feasible this would be in locations with fewer resources should be discussed. Many areas, including ours, are experiencing a shortage of pharmacists.

3.4 Author Response

Thank you we agree the intervention was resource heavy in the discussion we now highlight the role of pharmacy technicians and suggest further economic evaluation:

Pharmacists in our study had medication review responsibilities only for between 9 and 24 residents, so allowing them dedicated time to come to understand those resident's individual needs. An economic evaluation would also indicate if this level of pharmacist input would be cost effective in practice. It may be that some tasks could be allocated to pharmacy technicians.

5. As the authors indicate, absence of residents' (and their relatives') views is a major limitation. How did funded patient and public involvement compensate for this?

3.5 Author Response

We agree the absence of the resident and their relatives voice is a limitation. We now include a sentence to this effect in the discussion

We tried to gain access to residents and relatives for interview in the process evaluation through the care home manager. This gatekeeper route was not successful, but we did not have ethical approval to access personal addresses so could not recruit directly. We had one short interview with a resident. We had three expressions of interest, but this was towards the end of the study and COVID regulations prevented us completing the interviews.

We have added a sentence explaining the characteristic of the public members

Public members had experience of having relatives receiving polypharmacy and of working in care homes; this enabled the research team to have confidence in their interpretations and to consider in greater detail potential resident focused benefits.

6 18 of 25 triads participated. Why did 7 decline the intervention or the interviews? Were the trial outcomes in these 7 any different? Was there any volunteer bias in care home recruitment? E.g. in the deprivation level of the location of the care home?

3.6 Author response

We have improved the reporting of non-sampled triads in the results page 7

There was no expression of interest from any stakeholder in seven triads. In three of these triads the PIPs did not deliver the intervention; in the remaining 4 triads the demographics and trial outcomes were similar to the interview sample.

We provide further explanation here on why we have confidence in this thematic analysis even though 4 triads who delivered the intervention were not sampled for interview. During the data analysis for the process evaluation, we mapped triad characteristics especially PIP demographics against accounts of deprescribing the activity most likely to reduce falls. In responses to your query, we looked back over this mapping and found that there were no differences in outcome across the sampled and non sampled triads. In the 4 non-sampled triads who delivered the intervention 2 PIPs were employed by GPs and 2 were not. The recent paper from our PhD student indicates that PIP employment might be a factors. Also during this earlier mapping, we explored IMD and there was sufficient variety in the interview sample to suggest we had a robust heterogenous sample.

7. The interview schedule, coding tree, and a sample of text coding should be appended.

3.7 Author Response

Thank you for this suggested we now include these as supplementary files

3. 8aMinor points. p.13, line 3, ‘dry skin’ should not be described as a ‘minor complaint’. Changed to condition

3.8aAuthor response Thank you for this close reading ‘Minor complaint’ is changed to ‘condition’

3.8b The terms ‘drug’ and ‘medication’ appear to be used interchangeably.

3.8b Author response

We have revised the manuscript for consistency in use of terms Medicine is used to refer to a prescribed drugs/medicines. Medication review or SRM is used to refer to the process of looking at a person’s medicines to see if changes are needed. An explanation of SMRs is proved in the introduction. Drug is used within the context of drug burden and an explanation of the Drug Burden Index is give in the introduction. The term medicine-related burden is used where burden is not purely due to a medicine’s chemical formulation.

3.8c A careful copy edit is needed.

3.8c Author responses. we have completed a proofread and made minor changes

3.8dPlease review absence of possessive apostrophe line 16/17

PIPS or PIPs? Suggest consistency.

3.8d Author Response

Unless the text refers to a single pharmacist independent prescriber (PIP) we have used the plural for ease of reading PIPs. For example page 10 a single GP referred to a single PIP.

VERSION 2 – REVIEW

REVIEWER	Jordan, Susan Swansea University, Nursing
REVIEW RETURNED	09-Mar-2023

GENERAL COMMENTS	A qualitative understanding of outcomes when pharmacist independent prescribers lead on medicine management in older people’s care homes Thank you for the opportunity to re-read this work. The paper is largely unchanged from the previous submission. I am not sure where earlier questions have been addressed: volunteer bias; costs of the intervention; relevance post-pandemic; drawing on existing literature. The new conclusion in the abstract relates to safety, but the results do not report safety outcomes – just respondents’ views.
--

	As reviewer 1 states, negative comments have been down-played. Now that we know that this large RCT did not demonstrate a difference in trial arms, these might offer useful insights into the absence of a clinical gain from the intervention. Table 1 should include the deprivation quintile of the care home, and the number of GPs in each practice. It is difficult to interpret data on numbers of patients, without numbers of doctors. The coding framework is very useful, and it would be good if space could be found to include this into the main paper. It shows that the responses coded were from a minority of the 38 participants. This should be commented. It is a shame that residents' families did not agree to be interviewed, but this is now acknowledged as a limitation. The paper reads well, but careful copy editing is needed.
--	---

VERSION 2 – AUTHOR RESPONSE

Reviewer: 3 Comments to the Author: A qualitative understanding of outcomes when pharmacist independent prescribers lead on medicine management in older people's care homes Thank you for the opportunity to re-read this work.

Point 1 The paper is largely unchanged from the previous submission. I am not sure where earlier questions have been addressed: volunteer bias; costs of the intervention; relevance post-pandemic; drawing on existing literature.

Response 1 We apologise as on reviewing we can see that these were not fully covered. These points are now added to the discussion to this version under a new sub heading strengths and limitations

Volunteer bias: now reported in limitation along with the limitations of spread of responses as evidenced in coding framework

A further point to note is that the sample, while appearing representative of the main trail sample, consist of those who volunteered for the additional process evaluation interview so may not represent the views of those who declined. This point along with the limited number of participants represented in the coding of a small number of codes means that the transferability of the data needs to be considered with caution. ~~Therefore a limitation of this research is that~~ as the results provide insights and understandings are from only a very specific group of HSCPs involved in supporting older resident care homes.

Costs of intervention: The economic evaluation did not demonstrate that the intervention was cost-effective when using QALYs as the outcome measure and will shortly be published as part of the final NIHR report'. Therefore in the paper I have added a sentence in the discussion stating more clearly that further economic evaluation would be required.

Further economic evaluation would be required to examine if this level of pharmacist input would be cost effective in practice. It may be that some tasks could be allocated to the developing role of pharmacy technicians (43).

Relevance post pandemic: we now reference that much clinical care has moved online, although anecdotally this is shifting again to in person page 16 now reads

A key aspect of the role was the presence of the pharmacy within the care home, the move to more 'on-line' working since the COVID-19 pandemic may make this aspect of the intervention more difficult to implement.

Drawing on existing literature: we are pleased to have the chance to revisit new literature in this area. In the discussion we include reference to Lexow et al 2022[who also found pharmacist review reduced drug relation problems for care home residents and work by Roughead et al 2022 which also failed to find significant change in falls in care home residents.

We reference the emerging role of clinical pharmacist technicians (consensus work by Street et al., 2023) as the role needs to be held under review as we identified pharmacist independent Prescribers were undertaking activity that would fall under a clinical pharmacist technician role, meaning the intervention may be cost effect in care homes

Point 2 The new conclusion in the abstract relates to safety, but the results do not report safety outcomes – just respondents' views.

Response 2 the abstract is changed now reads

This study suggests that pharmacist independent prescribers in primary care working within the multi-disciplinary team can safely manage care home residents' medicines leading to subjective improvements in residents' well-being and medicine management procedures.

Point 3 As reviewer 1 states, negative comments have been down-played. Now that we know that this large RCT did not demonstrate a difference in trial arms, these might offer useful insights into the absence of a clinical gain from the intervention.

Responses 3 We suggest there is a balance of views expressed in the results as previously stated by review 3 and following small changes at last review. We fully agree that as the main trial paper is now published there is a need to refer readers to this we have added to page 16, along with the comment that there needs to be economic evaluation

The CHIPPS RCT found the fall rate risk ratio for the intervention group compared with the control group was not significant. However, the Drug Burden Index outcome significantly favoured the intervention

Point 4 Table 1 should include the deprivation quintile of the care home, and the number of GPs in each practice. It is difficult to interpret data on numbers of patients, without numbers of doctors.

Response 4 We have amended the table to include IMD for care homes and provide detail on location of GP practice. We do not consider it appropriate to add number of GPs in practice for three reasons: 1) the intervention triad working relationship was between 1 GP and 1 pharmacist, 2) many GP practices have locum and part time staff so numbers of GP vary day to day; 3) if available including exact number of GPs increases risk of data becoming identifiable..

Point 5 The coding framework is very useful, and it would be good if space could be found to include this into the main paper. It shows that the responses coded were from a minority of the 38 participants. This should be commented.

Responses 5 It is not possible to include in main paper but we refer to the limitation of the number of respondents in some codes on page 17 within the subsection newly titled Strengths and limitations

This point along with the limited number of participants represented in the coding of a small number of codes means that the transferability of the data needs to be considered with caution. ~~Therefore a limitation of this research is that~~ as the results provide insights and

understandings are from only a very specific group of HSCPs involved in supporting older resident care homes

Point 6 It is a shame that residents' families did not agree to be interviewed, but this is now acknowledged as a limitation. The paper reads well, but careful copy editing is needed. Reviewer: 3
Competing interests of Reviewer: Our group works in the area of medicines optimisation.

Response 6 thank you for your comments and help in improving this paper. We hope you concur all remaining suggestions are addressed.